# Successor Uncertainties: exploration and uncertainty in temporal difference learning

## Abstract

We consider the problem of balancing exploration and exploitation in sequential decision making problems. This trade-off naturally lends itself to probabilistic modelling. For a probabilistic approach to be effective, considering uncertainty about all *immediate* and *long-term* consequences of agent's actions is vital. An estimate of such uncertainty can be leveraged to guide exploration even in situations where the agent needs to perform a potentially long sequence of actions before reaching an under-explored area of the environment. This observation was made by the authors of the Uncertainty Bellman Equation model (O'Donoghue et al., 2018), which explicitly considers full marginal uncertainty for each decision the agent faces. However, their model still considers a fully factorised posterior over the consequences of each action, meaning that dependencies vital for correlated long-term exploration are ignored. We go a step beyond and develop *Successor Uncertainties*, a probabilistic model for the state-action value function of a Markov Decision Process with a non-factorised covariance. We demonstrate how this leads to greatly improved performance on classic tabular exploration benchmarks and show strong performance of our method on a subset of Atari baselines. Overall, Successor Uncertainties provides a better probabilistic model for temporal difference learning at a similar computational cost to its predecessors.

## 1 Introduction

We consider a sequential decision making problem in which an agent interacts with an unknown environment, modelled as a Markov Decision Process (MDP). The agent's goal is to learn a policy that maximises the expected cumulative reward while keeping the total number of interactions with the environment low. To do so, the agent has to balance exploration of the environment and exploitation of the information it has already acquired.

Because the agent learns sequentially through interaction with a potentially stochastic environment, the consequences of each action are not necessarily known ahead of time and this complicates striking the optimal trade-off between exploration and exploitation. A particular strategy to address this issue is maintaining a probabilistic model that reflects agent's current knowledge, or lack thereof, about the future effects of each action for any given state. Vitally, the model must incorporate uncertainty about not only the immediate but also the downstream effects of agent's actions, and reflect the varying degree of uncertainty associated with different sequences of states encountered and actions taken by the agent.

We will therefore focus on two concepts we believe to be crucial for effective exploration under uncertainty: *propagation of uncertainty* and *incorporation of dependencies*. We will say that a probabilistic model of the Q function *propagates uncertainty* whenever the degree of uncertainty over the Q value for each state-action pair is determined by the cumulative uncertainty of all future state-actions the agent may encounter. Notice that for any joint distribution of the Q function that propagates uncertainty, the model obtained by taking the product of its marginal distributions over Q values also propagates uncertainty. In other words, propagation of uncertainty does not guarantee that any dependencies between individual Q values, like the one implied by the Bellman equation, are captured by the model.

To this end, we will say that a model *incorporates dependencies* if the dependency structure of individual Q values obeys the Bellman equation. Further details are provided in section 3.

We illustrate importance of propagation of uncertainties and incorporation of dependencies in figure 1. The agent is faced with a sequence of binary UP/DOWN choices and receives a positive reward if and only if it executes a particularly long sequence of uninterrupted UP actions. In such a scenario, purely random exploration is extremely ineffective. Similarly, exploration which considers just immediate consequences of actions also performs poorly — shortsightedly reducing one-step uncertainty leads to exploration equivalent in expectation to that of a uniform exploration policy (more on this in section 6). In contrast, a model which propagates uncertainty and incorporates dependencies can be used to guide the agent into under-explored areas irrespective of their distance from the current state, resulting in more effective exploration of the environment. Whilst simple, this MDP captures the key features of classic exploration benchmarks like Montezuma's Revenge (Bellemare et al., 2013), where exploration is hindered by sparse reward and low connectivity of the state-space.

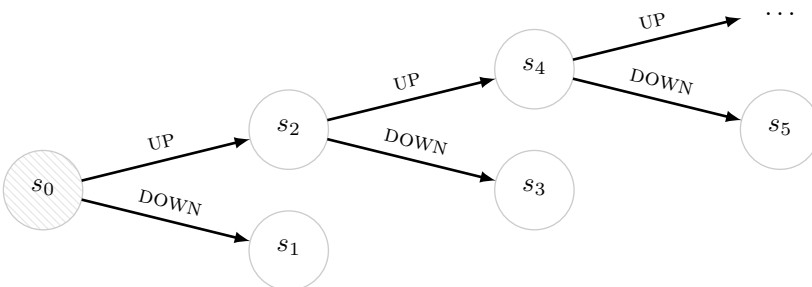

Figure 1: Binary TREE MDP; in each state, agent moves either UP or DOWN. States with odd indices are terminal. Reward is only obtained after a large fixed number of UP actions.

The main contributions of our work are summarised below:

1. Identification of *propagation of uncertainty* and *incorporation of uncertainties* as desirable attributes of models used for probabilistic exploration;

2. Introduction of *Successor Uncertainties*, a probabilistic model which both propagates uncertainty and incorporates dependencies;

3. Empirical investigation demonstrating how lack of either *propagation of uncertainty* or *incorporation of dependencies* can lead to inefficient exploration on classic exploration benchmarks (Dearden et al., 1998). Experiments on the Atari benchmark suite (Bellemare et al., 2013) confirming that Successor Uncertainties can be scaled to work in complex domains.

## 2 BACKGROUND

Before proceeding, we briefly review *model-free reinforcement learning* and *Bayesian linear regression*. These are key concepts we shall use throughout the remainder of our paper.

**Markov Decision Process** We consider a finite MDP with a horizon $H \in \mathbb{N}$, defined on a probability space $(\Omega, \mathcal{F}, P)$. An MDP is a tuple $(\mathcal{S}, \mathcal{A}, \mathcal{T})$, where $\mathcal{S}$ is a finite state space, $\mathcal{A}$ a finite action space, and $\mathcal{T} : \mathcal{S} \times \mathcal{A} \to \mathcal{P}(\mathcal{S} \times \mathcal{R})$ a transition probability kernel with $\mathcal{R} \subset \mathbb{R}$ a bounded reward set, and $\mathcal{P}(\cdot)$ the set of probability measures on given space. For each time step $t \in \mathbb{N}$, the agent selects an action $A_t$ by sampling from a distribution specified by the policy $\pi_t : \mathcal{S} \to \mathcal{P}(\mathcal{A})$ for the current state $S_t$, and receives a new state and reward $(S_{t+1}, R_{t+1}) \sim \mathcal{T}(S_t, A_t)$. This gives rise to a random process $(S_t, A_t)_{t \geq 0}$, Markov with respect to its natural filtration $(\mathcal{F}_t)_{t \geq 0}$, and a sequence of random variables $(R_t)_{t \geq 1}$. The task of solving an MDP is that of finding an optimal policy $\pi = (\pi_t)_{0 \leq t \leq H}$ that maximises the expected return $J^\pi = \mathbb{E}[\sum_{\tau=0}^{H} \gamma^\tau R_{\tau+1}]$ where $\gamma \in [0, 1)$ is a discount factor.

**Model-free reinforcement learning** All exploration methods we consider repose on a *model-free* approach to solving MDPs. These approaches invariably use estimates of *state-action values* and the corresponding *Q function*, $Q_t^\pi = \mathbb{E}[\sum_{\tau=t}^H \gamma^{\tau-t} R_{\tau+1} \mid \mathcal{F}_t]$, to guide exploration. Because computation of $Q_t^\pi$ is usually intractable, it is estimated for each $(s, a) \in \mathcal{S} \times \mathcal{A}$ using a (stationary) parametric model $\hat{Q}^\pi \colon \mathcal{S} \times \mathcal{A} \to \mathbb{R}$. This model is trained to obey the *Bellman equation*, $Q_t^\pi = \mathbb{E}[R_{t+1} \mid \mathcal{F}_t] + \gamma \mathbb{E}[Q_{t+1}^\pi \mid \mathcal{F}_t]$, by minimising a monotonically increasing function of the *temporal difference error* $\ell_{\mathrm{td}} \colon (s, a, s', r, \pi) \mapsto |\hat{Q}^\pi(s, a) - r - \gamma \mathbb{E}_{a' \sim \pi(s')}[\hat{Q}^\pi(s', a')]|$ on the data collected through exploration. The result of this procedure is an estimate of the *optimal policy* defined by greedy behaviour with respect to $\hat{Q}_t^\pi$, that is $\pi^\star \colon s \mapsto |A|^{-1} \delta_A$ with $A = \{a \colon \hat{Q}^\pi(s, a) = \max_{a' \in \mathcal{A}} \hat{Q}^\pi(s, a')\}$ and $|A|$ its cardinality.

**Bayesian linear regression** The goal of linear regression is to fit a model of the form $r = \langle \phi, w \rangle$ using a data set $D = \{(\phi_i, r_i)\}_{i=0}^t$. We will treat estimation of $w$ in a fully Bayesian manner, assuming a multivariate normal prior $P_w = \mathcal{N}(0, \Sigma_p)$ for $w$ and a Gaussian likelihood $P_{r|\phi,w} = \mathcal{N}(\langle \phi, w \rangle, \beta)$. By application of the Bayes' rule, the posterior over the weights given $D$ is then $P_{w|D} = \mathcal{N}(\mu_w, \Sigma_w)$. Here $\Sigma_w = (\Sigma_p + \beta^{-1} \sum_{i=0}^t \phi_i \phi_i^T)^{-1}$ and $\mu_w = \Sigma_w(\sum_{i=0}^t r_i \phi_i)$. Both $\mu_w$ and $\Sigma_w$ can be computed in an online fashion.

## 3 Probabilistic exploration in reinforcement learning

One of the most popular approaches to probabilistic exploration is *posterior sampling* (Dearden et al., 1998; Strens, 2000), also known as *Thompson sampling* (Thompson, 1933). The key feature of this method is that a distribution $P_{\hat{Q}^\pi}$ over $\hat{Q}^\pi$ is maintained instead of a point estimate. Exploration then proceeds by iterating the following steps: (i) sample $\hat{Q}^\pi \sim P_{\hat{Q}^\pi}$; (ii) explore the environment using the greedy policy with respect to the sampled $\hat{Q}^\pi$; (iii) update $P_{\hat{Q}^\pi}$ based on the collected data.

Effectiveness of posterior sampling is determined by properties of $P_{\hat{Q}^\pi}$. We will focus on two properties that we believe to be of vital importance.

> *1. Propagation of uncertainty:* the *marginal* distribution over $\hat{Q}^\pi(s, a)$ must reflect the total downstream uncertainty about all $\hat{Q}^\pi(s', a')$ that may be encountered when taking action $a$ in state $s$ and then following policy $\pi$.

Propagation of uncertainty ensures that uncertainty at any step is connected to *expected* uncertainties at subsequent time-steps (O'Donoghue et al., 2018). It is sufficient for approaches based on marginal statistics, for example upper confidence bound exploration (Auer et al., 2002), but does not guarantee coherence of individual samples.

> *2. Incorporation of dependencies:* the set of random variables $\{\hat{Q}^\pi(s, a)\}_{s \in \mathcal{S}, a \in \mathcal{A}}$ must (almost surely) obey dependencies specified by the Bellman equation.

Posterior sampling uses greedy policy with respect to *a single sample* from $P_{\hat{Q}^\pi}$. It is thus crucial that the value at any time-step is connected to the expected value at subsequent time-steps for *every* such sample, not only in expectation. If a sample which does not satisfy the Bellman equation is used, individual sampled Q values may carry conflicting information at each time-step. Importantly for our discussion later, a model $P_{\hat{Q}^\pi}$ propagates uncertainty if it incorporates dependencies but the converse is not necessarily true.

## 4 Uncertainty propagation through Successor Features

We present a new method, *Successsor Uncertainties*, which employs a probabilistic model $P_{\hat{Q}^\pi}$ that does not assume a factorisation of the posterior over the state-action space. The model both propagates uncertainties and incorporates dependencies, and thus leads to temporally correlated exploration when combined with posterior sampling.

## 4.1 The Successor Uncertainty model

Suppose that an embedding $\phi\colon \mathcal{S} \times \mathcal{A} \to \mathbb{R}^d_+$ exists, $d \in \mathbb{N}$, such that $||\phi||_2 = 1$ and $\mathbb{E}\left[R_{t+1} \mid \mathcal{F}_t\right] = \langle \phi_t, w \rangle$ for some $w$ in $\mathbb{R}^d$ and $\phi_t = \phi(S_t, A_t)$. Then

$$Q^\pi_t = \mathbb{E}\left[\sum_{\tau=t}^{H} \gamma^{\tau-t} R_{\tau+1} \mid \mathcal{F}_t\right] = \mathbb{E}\left[\sum_{\tau=t}^{H} \gamma^{\tau-t} \langle \phi_\tau, w \rangle \mid \mathcal{F}_t\right] = \left\langle \mathbb{E}_\pi\left[\sum_{\tau=t}^{H} \gamma^{\tau-t} \phi_\tau \mid \mathcal{F}_t\right], w \right\rangle,$$

where the second equality follows from the tower property of conditional expectation and the third is a simple application of the dominated convergence theorem. We define $\psi^\pi_t = \mathbb{E}[\sum_{\tau=t}^{H} \gamma^{\tau-t} \phi_\tau \mid \mathcal{F}_t]$ which allows us to express the Q function succinctly in terms of this quantity as $Q^\pi_t = \langle \psi^\pi_t, w \rangle$. The random vectors $\psi^\pi_t$ are known as *successor features* in the literature (Dayan, 1993; Barreto et al., 2017), and correspond to the (discounted) expected future occurrence of each feature when following a given policy $\pi$.

We develop the Successor Uncertainties model for $Q^\pi_t$, denoted by $\hat{Q}^\pi_{\mathrm{SF}}$, by modelling the quantities $w$ and $\psi^\pi_t$. We use a stationary estimator of the successor features $\hat{\psi}^\pi$ which will be obtained by noting that $\psi^\pi_t = \phi_t + \gamma \mathbb{E}[\psi^\pi_{t+1} \mid \mathcal{F}_t]$, meaning any standard temporal difference approach applies. We perform Bayesian linear regression to infer a distribution over $w$ using a Gaussian prior $\mathcal{N}(0, \alpha I)$ and likelihood $\mathcal{N}(\langle \phi, w \rangle, \beta)$. This makes the computation of the posterior over $w$ given $D$, $P_{w|D} = \mathcal{N}(\mu_w, \Sigma_w)$, with $D$ the set of all so far observed states and rewards, analytically tractable. Finally, stacking all successor feature estimates into a matrix $\hat{\Psi}^\pi = [\hat{\psi}^\pi(s, a)]^\top_{s \in \mathcal{S}, a \in \mathcal{A}}$, the implied distribution over $\hat{Q}^\pi_{\mathrm{SF}}$ can be shown to be

$$\mathrm{Law}(\hat{Q}^\pi_{\mathrm{SF}}) = \mathcal{N}(\hat{\Psi}^\pi \mu_w, \hat{\Psi}^\pi \Sigma_w (\hat{\Psi}^\pi)^\top),$$

highlighting the dependence between individual $\hat{Q}^\pi_{\mathrm{SF}}(s, a)$ introduced by our construction.

A limitation of Successor Uncertainties is that only the rewards are explicitly treated in a probabilistic way, whereas a point estimate is used for the transition model in construction of $\hat{\psi}^\pi$. This issue may be partially alleviated following an argument analogous to the one made by O'Donoghue et al. (2018). Specifically, if we are in a tabular setting and $\phi(s, a)$ are one-hot encoded representations of individual state-actions, the posterior variance $\Sigma_w$ will be diagonal with $\beta(n_{sa} + \frac{\beta}{\alpha})^{-1}$ for its non-zero entries, where $n_{sa}$ the number of times the tuple $(s, a)$ has been observed. Combined with the above derived $\mathrm{Law}(\hat{Q}^\pi_{\mathrm{SF}})$, the marginal variance over $\hat{Q}^\pi_{\mathrm{SF}}(s, a)$ will be proportional to $\beta n^{-1}_{sa}$ which will decrease at the same rate as if we had incorporated a Dirichlet model over the transitions (O'Donoghue et al., 2018). Thus, if $\alpha = \beta$ and $\beta$ is treated as a hyperparameter (as in our experiments), setting $\beta$ sufficiently high can adjust our estimate for the uncertainty about the transition model. As in (O'Donoghue et al., 2018), a similar argument can be made in terms of *pseudo-counts* (Bellemare et al., 2016) for linear Q function models.

Finally, our derivations show that $\psi^\pi$ needs to be relearnt for each policy $\pi$ which would generally require a significant amount of computational resources. To address this issue, we instead learn $\psi^{\bar{\pi}^\star}$ where $\bar{\pi}^\star = \int \pi^\star_w dP_{w|D}$ is the average over policies $\pi^\star_w$; here $\pi^\star_w$ is the greedy policy with respect to the particular Q function sample determined by $w$. This approximation reduces coherence of the dependencies within each Q function sample, but allows the algorithm to scale to large applications.

## 4.2 Successor Uncertainty with Neural Network embeddings

One of the main assumptions we made is that the embedding function $\phi$ is known a priori. This section considers the scenario where $\phi$ has to be learnt jointly with other parameters.

To distinguish from the fixed $\phi$, we use $\hat{\phi}\colon \mathcal{S} \times \mathcal{A} \to \mathbb{R}^d_+$ to refer to the fitted model. Denoting $(s_t, a_t)$ the state-action observed at step $t$, $\hat{\phi}_t = \hat{\phi}(s_t, a_t)$ and $\hat{\psi}^\pi_t = \hat{\psi}^\pi(s_t, a_t)$, we propose to learn $\hat{\phi}$ and $\hat{\psi}^\pi$ jointly by enforcing the known relationships between $\phi_t$, $\psi^\pi_t$ and $\mathbb{E}[R_{t+1}|\mathcal{F}_t]$:

$$\text{minimise} \quad \|\langle \hat{w}, \hat{\psi}^\pi_t \rangle - \gamma(\langle \hat{w}, \hat{\psi}^\pi_{t+1} \rangle)^- - r_{t+1}\|^2_2 + \|\langle \hat{w}, \hat{\phi}_t \rangle - r_{t+1}\|^2_2 \quad \text{w.r.t. } \hat{\phi}_t, \hat{\psi}^\pi_t, \hat{w} \quad (1)$$

$$\text{subject to} \quad \hat{\psi}^\pi_t = \hat{\phi}_t + \gamma \mathbb{E}_{\pi(s_{t+1})}[\hat{\psi}^\pi(s_{t+1}, a'_{t+1})], \quad (2)$$

$$\hat{\phi}_t, \hat{\psi}^\pi_t \geq 0 \text{ elementwise and } \|\hat{\phi}_t\|_2 = 1, \quad (3)$$

where $r_{t+1}$ is the reward observed after taking action $a_t$ in state $s_t$, $(\cdot)^-$ means that the quantity $(\cdot)$ is treated as fixed, and $\hat{w}$ is an auxiliary variable which can be either optimised or replaced with $w$ and integrated out with respect to $\mathcal{N}(\mu_w, \Sigma_w)$. Please note that the problem is stated in terms of single step $t$ to reduce clutter, but average over mini-batches sampled from a replay buffer will be optimised.

To simplify the optimisation, we will only softly enforce the constraint in equation (2) by adding a regularisation term $\|\hat{\psi}_t^\pi - (\hat{\phi}_t + \gamma \, \mathbb{E}_{\pi(s_{t+1})}[\hat{\psi}^\pi(s_{t+1}, a'_{t+1})])^-\|_2^2$ to the objective in equation (1). Empirically, the weight of the regularisation term can be set to one for all problems, eliminating what would otherwise be a hyperparameter.

Notice that the first term in equation (1) is the standard temporal difference learning loss. Now if a solution achieving zero temporal difference loss exists, then setting each $\hat{\phi}_t = \hat{\psi}_t^\pi - \gamma \hat{\psi}_{t+1}^\pi$ for a particular policy $\pi$ will make the second term in equation (1) zero. In other words, the additional terms in equations (1) and (2) do *not* affect the set of parameters that solve the Q function estimation problem. Because the non-negativity condition is trivially satisfied with ReLU activations, the only change with potentially significant impact on optima is the $\|\hat{\phi}_t\|_2 = 1$ restriction which impacts the norm of $\hat{\psi}_t$ through equation (2). In our experience, this constraint simplifies selection of the prior over $w$ as it keeps the scale of the inputs constant across all problems, increasing the robustness of our method.

We illustrate integration of neural network embeddings on our Atari architecture which is a modification of the DDQN model of Van Hasselt et al. (2016). As you can see in figure 2, Successor Uncertainties can be combined with existing reinforcement learning frameworks simply by including an additional neural network head for prediction of immediate rewards. The whole model is then optimised by minimising the above described relaxed version of the problem in equations (1-3), with the unit norm constraint on $\hat{\phi}$ enforced by explicit normalisation, and the non-negativity by using the ReLU activation.

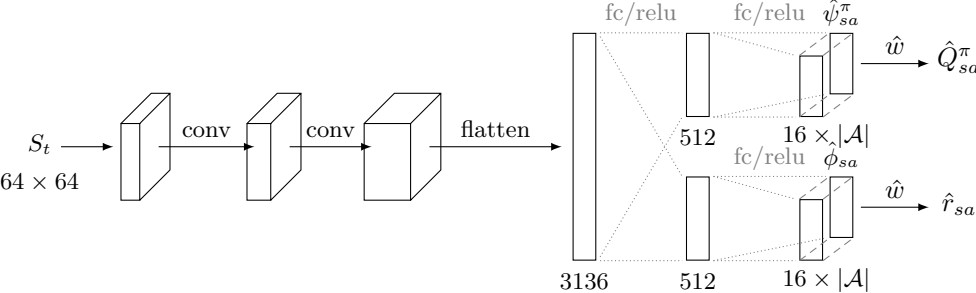

Figure 2: Architecture diagram for network used for Atari experiments.

We would like to emphasise that, unlike in previous work utilising successor embeddings (Kulkarni et al., 2016; Machado et al., 2017; 2018), our approach does not employ an auxiliary state reconstruction or state-transition prediction model in order to learn meaningful representations. This significantly reduces the implementation effort, and, importantly, makes comparison of Successor Uncertainties to other methods easier as state-prediction is in itself a very strong auxiliary loss even without improved exploration.

## 5 Related probabilistic exploration methods

Having introduced Successor Uncertainties, we now discuss several alternative probabilistic models of the Q function, focusing on if and how they propagate uncertainties and incorporate dependencies.

**Variational Q-learning** Deep Q-learning (Mnih et al., 2015; Watkins & Dayan, 1992) is a method which uses a deep neural network (Q-network) to estimate Q function of a particular exploration policy; the network is trained by minimising the temporal difference error on past observations seen by the agent. Some recent methods have attempted to replace

the deterministic Q-network with a Bayesian neural network (BNN) and apply posterior sampling using the Bayesian uncertainty estimates (for example Lipton et al., 2016; Gal, 2016). A common feature of these approaches is that the Q value *estimates* are treated as if they were real observations Q values in a regression problem. As shown by Riquelme et al. (2018), if the Q value estimates are inaccurate, which often happens before the environment is sufficiently well explored, variational Q-learning uncertainty estimates can be slow to converge and result in poor exploration.

What is more, uncertainty estimates of these methods rely only on the particular state-action pair $(s, a)$ under consideration. Whilst the temporal difference updates ensure that the Q value for action $a$ in state $s$ does reflect the rewards that may be obtained by taking action $a$ and following a given policy $\pi$ afterwards, *uncertainty* about those expected future rewards is not explicitly propagated.

**Bayesian linear Q-function prediction**   Linear Q function models, despite their simplicity (or perhaps because of it), have proven a good modelling choice in reinforcement learning (Levine et al., 2017; Osband et al., 2014; Lagoudakis & Parr, 2003). One of the possible explanations is that the tractability of the closed form updates often translates into lower variance and better calibrated uncertainty estimates as compared to variational approaches (Riquelme et al., 2018).

Bayesian linear Q function prediction models, such as Bayesian Deep Q-Networks (BDQN Azizzadenesheli et al., 2018) or Double Uncertain Value Networks (Moerland et al., 2017), utilise a standard Q-network: state $s \in \mathcal{S}$ is mapped to a state embedding $\phi_s^\pi$, which is then used together with a vector $u_a$ for each action $a \in \mathcal{A}$ to predict $\hat{Q}^\pi(s, a) = \langle u_a, \phi_s^\pi \rangle$. An auxiliary Bayesian linear model is then used to independently model uncertainty over each $u_a$. Analogously to Successor Uncertainties, the outputs of the Bayesian linear models define a distribution over Q functions $P_{\hat{Q}^\pi}$, which is used for exploration through posterior sampling. However, as in variational Q-learning, uncertainties are not explicitly propagated.

**Uncertainty Bellman Equation**   The Uncertainty Bellman Equation (UBE) model (O'Donoghue et al., 2018) introduces a dynamic programming rule for accumulating Q function uncertainty estimates such that they reflect expected uncertainty over all future consequences. The method performs very well on the Atari benchmark suite (Bellemare et al., 2013), achieving strong results on the Montezuma's Revenge exploration benchmark.

The UBE model assumes that the MDP is acyclic and thus that $R_i$ is independent of $R_j$ given $\pi$ and $\mathcal{T}$ for $j > i$. It further assumes that local uncertainty estimates associated with each state-action tuple are available. These are denoted $V_t$, and are estimated through the use of a Bayesian linear model in the form $\hat{V}: \mathcal{S} \times \mathcal{A} \to \mathbb{R}_+$, as within the Bayesian Deep Q-Networks method. UBE then defines the expected future uncertainty

$$U_t^\pi = \mathbb{E}\left[\sum_{\tau=t}^{H} \gamma^{2(\tau-t)} V_{t+1} \mid \mathcal{F}_t\right] = \mathbb{E}[V_{t+1} \mid \mathcal{F}_t] + \gamma^2 \mathbb{E}_\pi\left[U_{t+1}^\pi \mid \mathcal{F}_t\right],$$

where the second equality follows by the assumption of independent rewards. The latter expression for $U_t^\pi$ allows for efficient learning of a neural network estimator $\hat{U}: \mathcal{S} \times \mathcal{A} \to \mathbb{R}_+$ through the use of temporal difference updates.

In effect, UBE takes uncertainty estimates that do not propagate uncertainty, like those of the previously discussed methods, and forms a corrected estimator $\hat{U}^\pi$ which *does* reflect downstream uncertainty. However, as $\hat{U}^\pi$ specifies a marginal distribution over each $\hat{Q}^\pi(s, a)$, but not a joint distribution over $\hat{Q}^\pi$, it does not incorporate dependencies between different states and actions as is apparent from the discussion before equation (3) in (O'Donoghue et al., 2018) where the implicit posterior has diagonal covariance matrix which implies independence in the case of jointly multivariate normal variables.

## 6 Experiments

We present two sets of experiments. First, tabular experiments, where we provide an analysis of a general scenario within which Successor Uncertainties outperforms Bayesian Deep Q-Networks (BDQN) and the Uncertainty Bellman Equation (UBE). Second, we provide a limited set of Atari experiments, where we show that Successor Uncertainties can provide scores competitive with BDQN and UBE.

### 6.1 Tabular exploration problems

Our experiments are based on two problems, the binary decision tree introduces first presented in the introduction (figure 1) and the Chain problem, introduced in (Dearden et al., 1998). The chain is illustrated in figure 3. The version we use follows from (Osband et al., 2016), consisting of $L+2$ states and two actions corresponding to a move left and right along the chain respectively. The horizon is $L+9$ allowing a maximal reward of 1.0. The simple to discover sub-optimal solution 'left always' makes the problem challenging. These two problems isolate exploration, that is the acquisition of data informative about the MDP, from learning, i.e. the manner in which that data is then used to learn a Q function estimator.

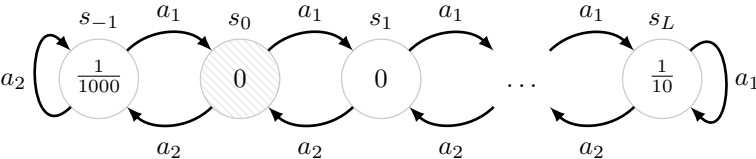

Figure 3: The CHAIN MDP illustrated

We present the return achieved by the three methods in these tabular problems within figure 4. The poor performance of BDQN follows from its lack of propagation of uncertainty. To understand the reason for the similarly poor performance of the Uncertainty Bellman Equation model, note that prior to the reward being reached, $\mu_{sa_1} = \mu_{s_2} = 0$ in expectation. Writing $\hat{Q}_{sa_i} \sim \mathcal{N}(0, \sigma_{sa_i})$ for the two actions UP and DOWN respectively, we see that $P_{A_0|S_0}(a_1 \mid s) = \mathbb{P}(\hat{Q}_{sa_1} - \hat{Q}_{sa_2} > 0) = 1/2$, irrespective of the variances $\sigma_{sa_i}^2$. Thus, on this task, posterior sampling with UBE uncertainty estimates results in exploration equivalent to that induced by a uniform random policy. Successor Uncertainties on the other hand distinctly outperforms both approaches. To see why, assume that $s_i$ is the furthest state seen thus far during exploration. Sample a weights vector $w \sim \mathcal{N}(\mu_w, \Sigma_w)$. Now if the entry $w_i$ associated with $s_i$ is high, not only does that increase the probability of selecting action UP in state $s_{i-1}$, but also for all states $s_0, \ldots, s_{i-1}$, as these contain a positive entry for $s_i$ in their successor representation $\psi^\pi$. Thus exploration utilising Successor Uncertainties is not equivalent to that of a uniform policy but instead explores in a correlated manner.

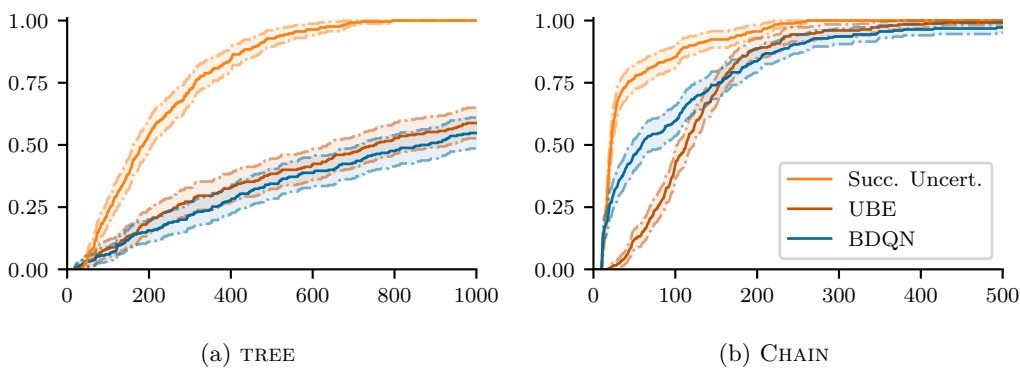

(a) TREE        (b) CHAIN

Figure 4: Test reward on TREE (left) and CHAIN (right), both with L=10. Episodes on $x$-axis. Average over 250 runs plotted; 95% CI shaded.

## 6.2 Atari experiments

We've run a series of small experiments (5M steps) on a subset of Atari games. Here, we test the performance Successor Uncertainties using embeddings learnt online through the constrained model-free reinforcement learning approach described in section 4.2.

We present our results in table 1, alongside results from related exploration methods. Test scores reported are obtained using the standard procedure set out in the seminal work on deep reinforcement learning for Atari (Mnih et al., 2015) and utilised subsequently (Van Hasselt et al., 2016; O'Donoghue et al., 2018). Games were selected by looking at training curves shown in (Azizzadenesheli et al., 2018) and the classification of game exploration difficulty in the appendix of (Ostrovski et al., 2017). Relatively difficult games where most of the improvement in score can be attained early within training were chosen, allowing for informative experiments with only a modest computational budget.

Table 1: Results for selected Atari games for our proposed method Successor Uncertainty, BDQN (Azizzadenesheli et al., 2018), UBE (O'Donoghue et al., 2018) and Double Deep Q-Network (DDQN) with $\epsilon$-greedy exploration (Van Hasselt et al., 2016). Numbers in parenthesis indicate the number of interactions with the environment undertaken by each method during training. Entries with dashes indicate no score was given in the source paper for that given game.

| Game | Method | | | |
| --- | --- | --- | --- | --- |
| | Succ. Uncert. (5M) | UBE (200M) | BDQN (30-40M) | DDQN (200M) |
| Bank Heist | **1,224** | 718 | 720 | 340 |
| Crazy Climber[1] | 111,770 | **132,998** | 124,000 | 101,874 |
| Enduro | **1,614** | 31 | 1,120 | 380 |
| Freeway | 28 | 0 | — | **32** |
| Ms Pacman[1] | 2,777 | 3,141 | — | **3,210** |
| Q*bert[1] | 14,290 (8M) | **16,772** | — | 14,875 |

Atari results are more difficult to interpret and analyse than results from tabular experiments. Performance on these games is highly related to how well the method interacts with the convolutional neural network model utilised, and is thus less clear in terms of an evaluation of exploration performance. Our results are comparable to those of other common exploration methods, showing that Successor Uncertainties can be scaled to work in complex domains where state-action embeddings have to be learnt in an online manner.

## 7 Discussion

We have introduced Successor Uncertainties, a probabilistic model for the Q function of an MDP that allows for efficient exploration at cost comparable to that of previous Q-learning methods. Our method separates local uncertainties from deterministic features learnt by a temporal difference approach. The benefit is that information is propagated within the state-action space of the MDP and statistical dependencies in the Q-function values are accounted for. Because of this, our method is a truly reinforcement learning exploration solution, rather than an adaptation of a contextual bandit method, like epsilon-greedy, to a reinforcement learning problem. Our results show that we compare favourably with UBE and BDQN, two related strong probabilistic modelling techniques, on a subset of Atari problems. We also test on classic exploration benchmarks, which assess the ability of methods to deal with sparse rewards and low connectivity of the state-space, both key aspects of exploration that strong methods must be capable of handling. Here, Successor Uncertainties significantly outperforms both BDQN and UBE, highlighting the importance of incorporating dependencies in probabilistic Q function models.

---

[1]Crazy Climber, Ms Pacman and Q*bert scores were steadily increasing as experiments ended.

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
