# OpenReview forum: "Successor Uncertainties: exploration and uncertainty in temporal difference learning"
_ICLR.cc/2019/Conference_

### Official Review · AnonReviewer2 · 2018-11-03
**Okay paper, but needs more substantiation**

**Rating:** 4
**Confidence:** 5

**Review:**

This paper tackles the classical exploration / exploitation problem in reinforcement learning.
The paper argues that it is necessary to propagate uncertainty correctly and argue that they can do so using the successor representation to compute the Bayesian posterior over Q-values conditioned on the data already observed.

Novelty:
This work is similar to “The uncertainty bellman equation” (UBE) (O’Donoghue et al. 2018) which adds a head to a regular DQN agent to predict a function u which is an upper bound of the variance of the posterior distribution over Q-values. The difference here is that the successor features are used here to predict Q-values and the function u.
The successor features can be seen as a discounted state occupancy of the current policy and carry information about the future. While the relation with the UBE is highlighted in section 4.6 an empirical evaluation between the two methods would also be needed.

Clarity: The method is detailed comparing to contextual bandit methods, the authors then argue that applying directly these methods to reinforcement learning case does not propagate uncertainty over several timesteps. While this indeed true it is misleading and imply that propagating uncertainty is not considered by current exploration methods in RL.

Soundness:
The method presented here is relatively reductive. The estimated posterior over the value function is correct only if the transition model P is already known, otherwise Equation 4 would also need to incorporate uncertainty over P. Similarly, as the authors point out, this doesn’t include the max operation. At best, we are learning a posterior over the value function for a fixed policy, for when only the reward is unknown.

As a whole, the authors argue that their method allows a better propagation of Q-values uncertainty but provide little theoretical or experimental evidence that would back this claim.

From a deep RL perspective, the features \phi^l only carry local information. The authors argue that this leads to more stable features as these feature do not depend on the current policy. However it also means that in a sparse reward setting the reward observed would be zero most of the time and no useful features would be learned. In practice methods using the successor representation usually share parts of the network with other tasks to improve representation learning (see e.g. Figure 1 of Machado et al., Eigenoption discovery through the deep successor representation, 2017, & also their 2018 paper).

Experiments:
The experiment are disappointing as they are only limited to tabular and deterministic problems. An obvious missing comparison is to UBE, at the minimum; and other “deep” algorithms such as BDQN, Bootstrap DQN, etc. Some of these algorithms have been shown to perform well on the Atari benchmark, and that seems like a reasonable point of comparison also.

The method also relies on knowing the successor features. While they can be learned easily in a tabular, deterministic MDP it is not clear how the posterior would behave in larger and/or non stochastic domains when it takes more time to learn these successor features.

Overall, I am not sure what I learned from reading this paper. While the idea of using the successor representation in exploration is interesting and has been considered recently, the method presented in this paper needs to be better justified and evaluated on more challenging tasks.


Minor comments
I would like to see a proof of Equation 4, which may be simple but is not immediate.

Some papers of relevance here:

An analysis of model-based Interval Estimation for Markov Decision
Processes, Strehl & Littman (2008)
(More) efficient reinforcement learning via posterior sampling. Osband et al (2013)
Count-Based Exploration with the Successor Representation, Machado et al. (2018)

---

> ### Author Response · Authors · 2018-12-07
> **Follow up on rebuttal**
>
> Dear AnonReviewer2,
>
> We’ve updated the paper, incorporating a great deal of your feedback and improving the experiments. To make it easy to check the changes, our rebuttal contains references to the sections where we address each point. When you have a moment to look over those changes, could you please let us know if these influence your rating of our paper?

---

### Official Review · AnonReviewer3 · 2018-11-06
**Exploration using successor features representations**

**Rating:** 5
**Confidence:** 4

**Review:**

The paper proposes an exploration approach (either based on posterior sampling or optimism) based on successor features representation. A high probability ellipsoid confidence set (defined by the Gram matrix \Sigma_t) is estimated based on linear regression (using some features \phi) to the immediate reward function. Now for any policy \pi, a confidence interval for the Q-value of any policy \pi can be derived by application of the \psi transformation, where \psi = expected sum of discounted future \phi under \pi. The algorithm selects action by posterior sampling (or UCB) in the \psi-space.

It would be interesting to see if this approach would converge to a good policy, maybe by doing a regret-based analysis.
Unfortunately there is no such analysis in the paper. However my main complaint is the soundness of the approach, for two reasons:
- First it is not clear that the uncertainty in the Q-values decreases with time. Indeed the uncertainty on Q^{\pi}(s,a) corresponds to the width of the confidence ellipsoid in the direction of the successor features \psi^{\pi}(s,a). However, although we know that the uncertainty shrinks in the directions of the features \phi_t (when action a_t is chosen in state s_t) because we do regression of the reward function, we do not have the same property for \psi_t, which defines the Q-function. And it is not obvious that the confidence set in the direction of \psi_t would shrink at all. Thus it could be the case that the uncertainty on the Q-values will never decrease.
- Second, since the successor features are learnt on-policy, the uncertainty on the Q-values (assuming we can estimate them) corresponds to a mixture of the policies which have been used in the past, but not to the policy that will be used from there on, because the policy is non stationary (since the uncertainty decreases as more information is collected). I would recommend to be very careful when defining and using the successor features by emphasizing the policy under which those features are defined.

So in the end the contribution is mainly algorithmic. However I find it hard to say anything about the proposed approach, whether it improves over previous ones or not, specially because the experiments are limited to toy problems. Theoretical analysis or more complex experiments would make the paper stronger.

---

> ### Author Response · Authors · 2018-12-07
> **Follow up on rebuttal**
>
> Dear AnonReviewer3,
>
> Thank you for your insightful comments. We have used these, and those of the other reviewers, to significantly improve the paper. We’ve included references in our rebuttal to each specific change we made. We now have a much stronger and better analysed set of experiments. Might you have a moment to look over these and let us know if this changes the rating you believe our paper deserves?

---

### Official Review · AnonReviewer1 · 2018-11-07
**Experimental results are too preliminary to assess the contribution**

**Rating:** 4
**Confidence:** 3

**Review:**

Summary/contribution:
This paper focuses on the problem of incorporating uncertainty into RL. The primary contribution is exploring the use of successor features for uncertainty prediction over Q-values. The proposed approach builds on O’Donoghue et al. (2018). The authors provide experiments that demonstrate improved performance on a chain MDP environment and a Tree environment.

Pros:
- I found this paper to be above average in terms of clarity.

Cons:
- The experiments evaluation is restricted to simplistic environments. The authors make an argument for why using successor features would be more "stable", but I found the experimental evidence to support this claim to be underwhelming.

Justification for rating:
This paper does a good job of articulating an interesting approach to the exploration problem using successor representations.  In the current form however, it is really lacking in experimental evidence to support the main claims/contributions. Currently the domains considered are somewhat toy which I do not find convincing enough to demonstrate the effectiveness of their approach.

Other:
- I would appreciate a discussion on the relationship to Machado et al. 2018 which explored count based exploration with successor representations.

---

> ### Author Response · Authors · 2018-12-07
> **Follow up on rebuttal**
>
> Dear AnonReviewer1,
>
> We’ve posted a revised version of the paper and a rebuttal. Specifically we’ve now got a stronger set of experiments, and a stronger argument for the reasons why our method may be preferred to O’Donoghue et al. (2018). Have you had a chance to have a look at these? If so, might these influence your rating of our paper?

---

### Author Response · Authors · 2018-11-27
**Rebuttal**

We thank the reviewers for their insightful comments. We have now modified the manuscript, and feel that we have addressed these thoroughly. The changes include: inclusion of new Atari experiments with promising results, better comparisons and analysis on existing experiments, and sections that address issues around soundness that were identified by the reviewers.

---

> ### Author Response · Authors · 2018-11-27
> **Conclusion**
>
> We have addressed the reviewers' concerns and improved the paper as follows:
>
> 1) We have performed experiments on tabular problems and on Atari, comparing to state-of-the-art methods: Uncertainty Bellman Equation (UBE) and Bayesian Deep Q Networks (BDQN).
>
> 2) Despite obtaining significant gains in the previous experiments, we have also indicated clearly why our method is prefered to UBE and BDQN: our sampled Q functions incorporate dependencies that UBE and BDQN ignore.
>
> 3) We have described in detail how to implement Successor Uncertainties with Neural Network embeddings by solving a constrained temporal difference learning problem, making our approach competitive with state-of-the-art methods.
>
> We would be very grateful if the reviewers could kindly inspect these changes and see if their opinion has changed.

---

> ### Author Response · Authors · 2018-11-27
> **On concerns regarding the soundness of the method**
>
> Our initially submitted manuscript was not very precise. We have now addressed this, and feel that the exposition is now much clearer. Here, we address the specific comments the reviewers had about the soundness of our method and refer to sections of the paper which go into more detail on the matter.
>
> AnonReviewer3 was concerned that even in the limit of observing an infinite stream of embeddings \phi_t, our uncertainty over w, and thus over Q, may fail to converge to zero. Our embeddings \phi_t are non-negative and their dimensionality d is finite (and small, d=16 in practice), and thus it is very unlikely, even without considering the learning process and objective, that \phi_t and \psi_t are orthogonal. This would be necessary for Q function uncertainties to not converge to 0. Furthermore, \psi_t are learnt by temporal difference methods (with convergence guarantees in the tabular/sufficient approximator capacity settings) and thus \psi_t will be in the linear span of \phi_t, so if uncertainty about all directions specified by \phi(s,a) decreases to zero, the same must hold for all \psi(s,a).
>
> AnonReviewer3 was very rightly concerned that, despite deriving that each set of weights w ought to have its own corresponding embedding \psi, we instead use an embedding \psi that is learnt for the marginal policy over w. We note that (at least assuming that the temporal difference updates converge) the embedding is for the average of current policies, and not older policies, as we employ off-policy learning. We now acknowledge this clearly within the paper as a key limitation of our method and explain that this is an approximation necessary for computational tractability; see section 4.1 (The Successor Uncertainty model), last paragraph. We hope that the current set of experiments is sufficient to reassure the reviewer that despite this approximation, the method has the potential to outperform other probabilistic exploration methods.
>
> AnonReviewer2 highlighted the inherent difficulties in learning successor features and expressed a degree of doubt about the scalability of the method. This is completely warranted, as the original manuscript did little to address how we would deal with scaling the proposed method. We now dedicate section 4.2 (Successor Uncertainty with Neural Network embeddings) to this matter. Rather than, as suggested by the reviewer, employing the common trick of using auxiliary transition predictions to improve embeddings, we instead pose the problem as constrained temporal difference learning for \hat{Q}. This formulation avoids the issues of sparsity of reward inherent to successor features, but allows us to make direct comparisons of performance with that of other methods, without the difficulty of having to control for the added state transition auxiliary task and retaining a very simple DDQN-like network architecture (as is done in MC Machado 2017 & MC Machado 2018). Moreover, we hope that the Atari experiments presented in section 6.2 (Atari experiments) convince the reviewers that the proposed approach to scale successor features is sound.
>
> Finally, we address AnonReviewer2’s comments on the correctness of our posterior. As is the case for many of current model-free RL frameworks, we do not explicitly model uncertainty over the transitions as scaling of probabilistic transition models to complicated problems like Atari remains a challenge. However, following an argument first made by O’Donoghue et al. (2018) for UBE, we show that it is possible to implicitly adjust for the additional uncertainty due to the unknown transition kernel. Specifically, by increasing our \beta hyperparameter, we can inflate the uncertainty estimate in a way that ensures that the total uncertainty is of the correct scale for tabular and linear Q value models. We discuss this in the penultimate paragraph of section 4.1 (The Successor Uncertainty model).

---

> ### Author Response · Authors · 2018-11-27
> **On Experiments**
>
> The main comments were related to the experimental evaluation of our method. Here, we made two major improvements to address this, which we describe next.
>
> First, we analysed our previous set of experiments more thoroughly. Section 6.1 (Tabular exploration problems) now contains a comparison with the Uncertainty Bellman Equation (UBE) on these problems. The comparison with Bayesian Deep Q-Networks (BDQN) has been marked more clearly --- it is no longer referred to as local uncertainties. Thus the section now contains the two most relevant baselines from the existing literature. We note that on these `simple’ tabular tasks, neither BDQN nor UBE induce exploration different from that of a uniform exploration policy (at least in expectation), as shown by our updated plots. In paragraph two of Section 6.1 we explain in detail why UBE and BDQN perform poorly on these problems, and the reasons for Successor Uncertainties’ better performance. In short, it comes down to the assumptions within BDQN and UBE that lead to a fully factorised posterior that does not incorporate dependencies between various states. Successor Uncertainties builds on those methods by including said dependencies.
>
> Second, we now include a small set of Atari experiments. These contain problems selected as being potentially hard (according to Ostrovski 2017) but where most of the improvement in score occurs within early stages of training, such that we can perform informative experiments with only modest compute resources at our disposal. Our method performs very well on a couple of the problems, beating UBE and BDQN within very few interactions with the environment. While the scale of these experiments is small, they show that Successor Uncertainties may be preferred to UBE and BDQN for some scenarios in non-tabular settings, and that the method itself transfers well to complex domains. We note that we do not use an auxiliary state-transition prediction loss, as is frequently done within other successor feature based work.

---

> ### Author Response · Authors · 2018-11-27
> **On relation to BDQN and UBE**
>
> We now address the claim by most reviewers regarding our poor motivation/distinction from UBE/BDQN. UBE improves on BDQN by (under their set of assumptions) properly modelling the marginal distribution of each state-action pair. Successor Uncertainties goes one step above and beyond in that it models the joint distribution over all state-action pairs without assuming independence of individual Q values. This is key to ensuring that the induced exploration is non-dithering, a quality that has been shown to be of great importance to effective exploration (see e.g. Osband 2016 paper on Bootstrapped DQN). We would like to ask that the reviewers kindly examine section 3 (Probabilistic exploration in reinforcement learning), where we discuss the concepts of propagation of uncertainty and incorporation of dependencies and how these relate to the quality of exploration. In section 5 (Related probabilistic exploration methods), we now make a clear argument for why at least one of these criteria is not fulfilled by both BDQN and UBE. We hope that this in-depth discussion will satisfy AnonReviewer2, who felt our initial exposition was misleading --- we agree that plenty of methods propagate uncertainty, but a large subsets of methods that scale to and perform well on Atari do not. This consideration of dependencies is the main contribution of the paper and clarifies why we are able to significantly outperform both BDQN and UBE in section 6.1 (Tabular exploration problems) on one several classical exploration problems (a version of the Chain problem features in Dearden’s Bayesian Q-learning paper, 1998).

---

### Meta-Review · Area_Chair1 · 2018-12-14

**Confidence:** 4
**Recommendation:** Reject

**Metareview:**

Pros:
- interesting algorithmic idea for using successor features to propagate uncertainty for use in epxloration
- clarity

Cons:
- moderate novelty
- initially only simplistic experiments (later complemented with Atari results)
- initially missing baseline comparisons
- no regret-based analysis
- questionable soundness because uncertainty is not guaranteed to go down

All the reviewers found the initial submission to be insufficient for acceptance, and the one reviewer who read the rebuttal/revision did not change their mind, despite the addition of some large-scale results (Atari).